# Radiomics of Biliary Tumors: A Systematic Review of Current Evidence

**DOI:** 10.3390/diagnostics12040826

**Published:** 2022-03-28

**Authors:** Francesco Fiz, Visala S Jayakody Arachchige, Matteo Gionso, Ilaria Pecorella, Apoorva Selvam, Dakota Russell Wheeler, Martina Sollini, Luca Viganò

**Affiliations:** 1Department of Nuclear Medicine, IRCCS Humanitas Research Hospital, 20089 Rozzano, Italy; francesco.fiz.nm@gmail.com (F.F.); martina.sollini@cancercenter.humanitas.it (M.S.); 2Department of Biomedical Sciences, Humanitas University, Pieve Emanuele, 20072 Milan, Italy; visalasubodhi.jayakodyarachchige@st.hunimed.eu (V.S.J.A.); matteo.gionso@st.hunimed.eu (M.G.); ilaria.pecorella@st.hunimed.eu (I.P.); apoorva.selvam@st.hunimed.eu (A.S.); dakotarussell.wheeler@st.hunimed.eu (D.R.W.); 3Division of Hepatobiliary and General Surgery, Department of Surgery, IRCCS Humanitas Research Hospital, 20089 Rozzano, Italy

**Keywords:** radiomics, texture analysis, biliary tumors, cholangiocarcinoma, prognosis

## Abstract

Biliary tumors are rare diseases with major clinical unmet needs. Standard imaging modalities provide neither a conclusive diagnosis nor robust biomarkers to drive treatment planning. In several neoplasms, texture analyses non-invasively unveiled tumor characteristics and aggressiveness. The present manuscript aims to summarize the available evidence about the role of radiomics in the management of biliary tumors. A systematic review was carried out through the most relevant databases. Original, English-language articles published before May 2021 were considered. Three main outcome measures were evaluated: prediction of pathology data; prediction of survival; and differential diagnosis. Twenty-seven studies, including a total of 3605 subjects, were identified. Mass-forming intrahepatic cholangiocarcinoma (ICC) was the subject of most studies (*n* = 21). Radiomics reliably predicted lymph node metastases (range, AUC = 0.729–0.900, accuracy = 0.69–0.83), tumor grading (AUC = 0.680–0.890, accuracy = 0.70–0.82), and survival (C-index = 0.673–0.889). Textural features allowed for the accurate differentiation of ICC from HCC, mixed HCC-ICC, and inflammatory masses (AUC > 0.800). For all endpoints (pathology/survival/diagnosis), the predictive/prognostic models combining radiomic and clinical data outperformed the standard clinical models. Some limitations must be acknowledged: all studies are retrospective; the analyzed imaging modalities and phases are heterogeneous; the adoption of signatures/scores limits the interpretability and applicability of results. In conclusion, radiomics may play a relevant role in the management of biliary tumors, from diagnosis to treatment planning. It provides new non-invasive biomarkers, which are complementary to the standard clinical biomarkers; however, further studies are needed for their implementation in clinical practice.

## 1. Introduction

Biliary tumors are rare neoplasms (incidence: 0.3–6 cases per 100,000 inhabitants per year) with a poor prognosis, i.e., a median survival shorter than one year if unresectable and a five-year survival rate ranging from 10 to 40% if undergoing complete resection [1,2,3]. They encompass a wide range of diseases—from intrahepatic cholangiocarcinoma to peri-hilar, gallbladder, and distal bile duct cancers—that have some distinctions: they differ in terms of epidemiology, cells of origin, genetics, clinical presentation and management, and prognosis [2]. Nevertheless, biliary tumors share some common characteristics [1,2]. In detail, complete surgery is their only potentially curative treatment, but most patients are diagnosed at an advanced stage and are unresectable [3,4,5,6]. Current systemic therapies have limited efficacy, while new promising targeted therapies and immunotherapies are under evaluation [1,7]. When managing biliary tumors, clinicians have to face some major unmet needs, first and foremost, the diagnosis. Standard imaging modalities cannot provide a conclusive differentiation between intrahepatic cholangiocarcinoma (ICC) and HCC, with few exceptions [8,9,10]. In patients with biliary stenoses, the differential diagnosis between inflammatory diseases and malignant diseases is uncertain and sometimes remains unsolved, even after percutaneous or endoluminal biopsies [11,12]. Second, in the current clinical practice, reliable non-invasive biomarkers that predict tumor aggressiveness and the benefits for patients from loco-regional treatment, namely surgery, are lacking [1,2,3,13,14]. The selection of candidates for resection relies on the morphology of the neoplasms during imaging, which scarcely correlates with their biology [3,15]. The most relevant prognosticators can be evaluated only a posteriori on the surgical specimen [4,13,14].

In recent years, new approaches to medical imaging have been developed, based on the analysis of the raw data of these images [16]. Among those, radiomics has been used most broadly due to its easy application and high reproducibility [17]. Radiomics is the high-throughput extraction of textural features from any imaging modality [17,18]. It relies on the quantitative analysis of the texture and features of a segmented region of interest through semi-automatic or automatic software. Radiomics translates imaging data into indices derived from grey-level histograms, shape analyses, and second-order matrices. The radiomic features of several tumors, based on patterns of pixels and voxels, have demonstrated a strong association with the pathology data and prognosis [19,20,21]. Even if the clinical applications of radiomic analyses are yet to be standardized, they could be a breakthrough in the near future.

The present systematic review aims to summarize and critically analyze the available evidence about the performance and potential clinical role of radiomics of biliary tumors. We will focus on differential diagnosis, non-invasive assessment of the pathology data, and prediction of long-term outcomes.

## 2. Materials and Methods

The authors performed a systematic search of articles pertinent to radiomics of biliary tumors in PubMed, Science Citation Index, Embase, clinicaltrial.gov databases, and web sources (Google Scholar). The following keywords were used: “cholangiocarcinoma” or “biliary tumor” combined with “radiomics” or “textural analysis” or “texture analysis” or “radiological features” or “voxel patterns”. All articles published before 30 April 2021 (including online first papers) were considered. The list of articles was screened for duplicates; if present, these were removed. Titles and abstracts of the identified articles were reviewed, and the following exclusion criteria were used: (a) full text in languages other than English; (b) topic out of the scope of the present review; (c) preclinical studies with no translational aspects (i.e., no experiments on human subjects); (d) phantom, analytical, or simulation studies; (e) case reports; (f) editorials, commentaries, and reviews; (g) conference proceedings. The studies including biliary tumors together with other tumors or mixed forms (e.g., HCC/ICC) were retained only if the results of radiomics for biliary tumors were separately provided. The full text of the selected articles was retrieved. If multiple publications from the same research group/institution with significant overlap in terms of aim(s) and population (>50%) were identified, only the study with the largest cohort was included. The reference list of the selected articles was screened for potentially eligible studies. The reference list of case reports, editorials, commentaries and reviews was also screened for the same purpose. Two authors (FF and MS) evaluated all manuscripts; in cases of discordance, a consensus was reached after discussion with a third author (LV).

The present project was registered in the PROSPERO database (https://www.crd.york.ac.uk/prospero/, temporary registration number 288589, last accessed 1 December 2021). The systematic review was carried out according to the PRISMA (Preferred Reporting Items for Systematic Reviews and Meta-Analyses) guidelines (checklist available as Appendix A) [22]. We did not formulate a PICO question because this was an explorative review about an innovative topic, and we planned to answer a general question (the current role of radiomics in biliary tumors) rather than a specific one.

### 2.1. Quality Assessment

The quality of each study was independently assessed by two reviewers (MS and FF) using the Quality Assessment of Diagnostic Accuracy Studies 2 (QUADAS-2) criteria [23]. As per QUADAS-2 scoring design, the risk of bias and the applicability were evaluated for the following domains: “patient selection”, “index test”, and “reference standard”. In the “flow and timing” domain, only the risk of bias was assessed. Based on the signaling questions and the match of the paper with the review purpose, we defined both the “risk of bias” and the “applicability” as “unclear” (0.5 points), “low” (0 points), or “high” (1 point). In case of discordance, a third reviewer (LV) assessed the paper while blinded to previous assessments to reach a final decision.

### 2.2. Data Collection

For each study, we collected the following data: (1) general features, including name of the authors, institution, country, journal, and year of publication; (2) type of biliary tumor, i.e., ICC, peri-hilar cholangiocarcinoma, gallbladder cancer, and extrahepatic bile duct tumor; (3) study characteristics, including study design (e.g., retrospective, prospective, etc.), sample size, funding, and conflicts of interest; (4) analysis of imaging modality (ultrasonography, computed tomography (CT), magnetic resonance imaging (MRI), or positron emission tomography-CT) and phases; (5) details of radiomics analysis, including the software and the extracted features (first- and second-order ones); (6) reference standard; (7) performances of radiomics. All data were cross-checked by at least three authors.

## 3. Results

After screening for duplicates and eligibility, 27 studies were included [24,25,26,27,28,29,30,31,32,33,34,35,36,37,38,39,40,41,42,43,44,45,46,47,48,49,50]. Figure 1 depicts the selection process. Most of the selected papers were published in the last 18 months (*n* = 18, 67%), and three-fourths (*n* = 20) were written by Chinese authors. All studies had a retrospective design. Regarding the imaging modality, eleven papers analyzed CT [24,25,32,37,39,42,43,44,46,48,49], ten MRI [27,28,30,33,34,35,38,40,45,47], three percutaneous ultrasonography (US) [29,36,50], and three multiple imaging modalities (CT and MRI) [26,31,41]. Most studies considered ICC (*n* = 21); the remaining studies analyzed the following tumor sites: extrahepatic biliary tumor (EBDT, *n* = 2) [27,30]; peri-hilar cholangiocarcinoma (*n* = 2) [43,45]; gallbladder tumor (*n* = 1) [50]; and mixed biliary tumors (*n* = 1) [48]. The mean number of patients was 133 (range, 17–345), and the treatment of the analyzed population was surgery in most studies. All studies but one considered both first and second order radiomic features, and 20 out of 27 (74%) studies provided a validation of the proposed predictive/prognostic model. The data from the papers are summarized in Table 1.

For the qualitative synthesis, three groups of studies were identified according to their subject: (1) prediction of pathology data; (2) prediction of survival; (3) differential diagnosis between malignant biliary tumors and other diseases. In the first group (prediction of pathology data), we included one paper analyzing the response to radioembolization [42]. In the second group (prediction of survival), we included one paper focusing on the prediction of futile surgery (explorative laparotomy or R2 surgery) [37].

Inter-observer and/or intra-observer agreements were mentioned in 13 studies (48%). Eight studies reported the values of the Cohen’s kappa or intraclass correlation coefficient with a value over 0.80 in all but one study (range, 0.53–0.98) [26,27,30,34,37,40,44,49]. In six papers, the values of agreement were used to select the radiomics features, with heterogeneous threshold values for acceptance (range, 0.55–0.90) [39,40,45,46,47,48].

Considering the quality assessment, the QUADAS-2 score demonstrated a low risk of bias and a good applicability across studies. The highest risk of bias was in the patient selection domain, the risk being high in fifteen studies (56%) and unclear in two (7%). No study had a high risk of bias in the index test or in the reference standard domains. Considering flow and timing, only two studies (7%) had a high risk of bias. The complete assessment is detailed in Table 2.

### 3.1. Prediction of Pathology Data

Fourteen papers analyzed pathology data: ten analyzed ICC [24,25,26,28,29,38,41,42,44,47], three analyzed EBDT [27,30,48], and one analyzed gallbladder cancers [48]. Six papers considered the lymph node metastases [25,27,30,38,41,48], four the tumor grading [26,27,29,30], two the microscopic vascular invasion [29,47], and six the other pathology data [24,26,28,29,40,44]. One study analyzed the capability of radiomics to predict the response to transarterial radioembolization (TARE) of ICC [42]. The reference standard was the surgical specimen in ten papers [25,26,27,28,30,38,40,41,44,48], the surgical specimen or tumor biopsy in two papers [29,47], the biopsy in one paper [24], and the radiological response evaluated at the post-treatment imaging in one paper (the paper focusing on TARE for ICC) [42]. Twelve out of the fourteen studies reported the predictive performances of a radiomic signature/score combining multiple textural features [25,27,28,29,30,38,40,41,42,44,47,48].

Radiomics predicted lymph nodes metastases with good performances, with AUC ranging from 0.729 to 0.900 and accuracy from 0.690 to 0.830 [27,30,38]. In a series of 155 ICC patients, Ji et al. observed that the radiomic model performed better than the clinical one (AUC = 0.871 vs. 0.722) [25]. Three studies, including the previous one (155 ICC, 148 ICC, and 274 mixed biliary tumors, respectively), considered a model combining radiomics with clinical and/or radiological data (Ca19-9 value + N status at CT [25], Ca 19-9 value + N status at MRI [38], or N status at CT [48]). All combined models had high AUC values (0.892, 0.870, and 0.800, respectively), and outperformed the pure clinical models (AUC = 0.722, 0.787, and 0.730) with a clinical net benefit.

Considering tumor grading, the results were discordant. King et al. observed no association [26], while three studies reported an adequate AUC and accuracy of radiomic scores (AUC = 0.680–0.890, accuracy = 0.70–0.82) [27,29,30]. Considering microvascular invasion, radiomics extracted from US had poor performances (AUC = 0.699) [29], while those extracted from MRI had much better results (126 ICC patients, AUC = 0.873 and accuracy = 0.850) [47]. Table 3 summarizes the models and results of the analyzed studies.

Radiomics have also been associated with IDH1 and IDH2 status, PD-1 and PDL1, VEGF, EGFR status, and immunophenotype [24,28,29,40,44]. Data are summarized in Appendix A. One Italian study [42] reported that the textural features extracted from the arterial phase (mean of grey levels and GLCM homogeneity) and delayed phase (skewness, kurtosis, and GLCM dissimilarity) of pre-TARE CT may predict the response to treatment with a good performances (AUC = 0.896).

### 3.2. Prediction of Survival

Thirteen studies analyzed the role of radiomics in predicting the survival of patients with biliary tumors. Of those, eight focused on overall survival [25,26,28,40,42,45,48,49], five on recurrence-free survival [25,26,42,48,49], and four on early recurrence risk [34,35,41,43]. One additional paper analyzed the capability of radiomics to predict futile surgery, i.e., R2 resection or explorative laparotomy in candidates to surgery [37]. All studies analyzed the performances of a radiomic signature/score combining multiple textural features into a single parameter. Seven papers demonstrated that radiomics leads to a clinical net benefit [25,28,34,35,43,48,49].

#### 3.2.1. Overall Survival

Of the eight studies about overall survival, six concerned ICC [25,26,28,40,42,49], one peri-hilar tumors [45], and one mixed biliary cancers [48]. Two papers failed to demonstrate an association between textural features and survival [26,42]. The remaining six papers demonstrated the prognostic value of radiomics but reported heterogeneous data: three studies detailed the C-index of the radiomic model, with intermediate performances reported in two studies (98 and 78 ICC patients with MRI-based data, C-index = 0.673 and 0.700, respectively) [28,40] and a high performance in one study (184 peri-hilar cholangiocarcinoma patients with MRI-based data, C-index = 0.877 in the training dataset and 0.756 in the validation one) [45]; two studies mentioned the radiomic signatures as independent prognosticators (hazard ratio) but did not detail the C-index of the model [25,48]; and one study reported neither the C-index nor the hazard ratio [49].

Three studies [28,45,49] compared the performances of the radiomic model with those of the models combining radiomics with clinical, radiological, and pathology data. The first two studies concerned ICC: (1) Zhang et al. (98 patients) demonstrated a preoperative model considering radiomic features extracted from MRI and imaging classification (parenchymal/ductal tumor); the CEA values had performances similar to the pathology-based model (PD-1 and PD-L1 expression, C-index = 0.721) [28]; (2) Park et al. (345 patients) reported that a clinical-radiological model based on tumor contour, tumor multiplicity, periductal tumor infiltration, extrahepatic organ invasion, and suspicion of LN metastases had similar performances to the radiomic study based on CT, but the combination of the two led to better performances (C-index = 0.680, +0.06) [49]. For peri-hilar tumors, Zhao et al. (184 patients) demonstrated that the addition of radiomics extracted from MRI to standard prognosticators (preoperative CEA values, radiological N stage, and invasion of hepatic artery at imaging) increased the C-index of the prognostic model (+0.12 in the training dataset and +0.06 in the validation one, C-index = 0.962 and 0.814, respectively) [45].

Table 4 summarizes the models and results of the analyzed studies.

#### 3.2.2. Recurrence-Free Survival

Of the five studies concerning recurrence-free survival (RFS), four considered ICC [25,26,42,49] and one gallbladder cancer and EBDT [48]. All studies reported an association between textural features and RFS, but only one study mentioned the C-index (0.690) [49]. The latter study (345 ICC patients, CT-based radiomics) compared the performances of the radiomic model with those of a combined clinical, radiological, and radiomic model [49]. As observed for overall survival, the clinical–radiological model (tumor contour, tumor multiplicity, periductal tumor infiltration and extrahepatic organ invasion, suspicion of LN metastases) had similar performances to the radiomic model, but the combination of the two led to an increase in the C-index value (+0.06 in both the training and validation datasets, C-index = 0.750 and 0.710, respectively). Table 4 summarizes the models and results of the analyzed studies.

#### 3.2.3. Early Recurrence

Of the four studies concerning early recurrence after the surgery of biliary tumors, three considered ICC [34,35,41] and one peri-hilar cholangiocarcinoma [43]. Early recurrence was defined as any recurrence that occurred within 2 years [34,35] or within 1 year from surgery [41,43]. In all studies, radiomics achieved good performances, with the AUC ranging from 0.742 to 0.889. Zhao L et al. (47 ICC patients) demonstrated that the clinical model (enhancement pattern at MRI + VEGFR) had the lowest performance (AUC = 0.798, accuracy = 0.702), the radiomic model had an intermediate performance (0.889 and 0.809, respectively), and the combination of the two has the highest performance (0.949 and 0.872, respectively) [34]. Two additional studies (209 ICC patients with MRI-based radiomics; and 274 peri-hilar cholangiocarcinoma patients with CT-based radiomics, respectively) reported higher performances of the model combining clinical and radiomic data in comparison with both the radiomic-only and clinical-only models (at external validation, AUC = 0.860 and 0.861, respectively) [35,43]. Table 5 summarizes the models and results of the analyzed studies.

### 3.3. Differential Diagnosis

Seven studies analyzed the role of radiomics in the differential diagnosis of biliary tumors from other diseases [31,32,33,36,39,46,50]. Six papers considered ICC [31,32,33,36,39,46], and one focused on gallbladder polyps [50]. Three studies analyzed CT [32,39,46], two percutaneous US [36,50], one MRI [33], and one both CT and MRI [31]. The standard for reference was the surgical specimen in five papers [32,36,39,46,50], and the liver biopsy in two [31,33].

Three studies focused on the differential diagnosis between HCC and ICC/mixed HCC–ICC [31,33,36], while three studies focused on the differential diagnosis between ICC and mixed HCC–ICC [31,32,36]. Radiomics achieved good performances with AUC ranging from 0.700 to 0.854. Two studies compared the performances of the radiomic model with those of a combined clinical, radiological, and radiomic model: (1) Lewis et al. (17 ICC vs. 36 HCC, MRI-based radiomics) reported an AUC increase of 0.100 in the differential diagnosis of ICC from HCC by adding radiomics to the patient’s gender and LI-RADS criteria (AUC = 0.900, accuracy = 0.800) [33]; (2) Zhang et al. (123 ICC vs. 66 mixed HCC–ICC, CT-based radiomics) reported an AUC increase of 0.140 for the differential diagnosis of mixed HCC-ICC by adding radiomics to the patients’ gender, AFP value, presence of hepatitis B viral infection, and intratumoral necrosis (AUC = 0.942) [32]. The combined model had superior performances to both clinical- and radiomic-only models.

Two papers focused on the differential diagnosis between ICC and benign diseases (intrahepatic lithiasis and inflammatory masses) [39,46]. They compared 53 and 61 patients affected by ICC with 78 and 84 patients affected by a benign disease, respectively. The radiomic features were extracted from CT in both studies and had good discriminatory capabilities (AUC over 0.800 both in training and validation datasets). However, the best performances were achieved when the radiomic model was combined with the clinical data (CEA and CA 19-9 in both studies, fever in one), outperforming the clinical- and radiomic-only models (AUC = 0.879 and 0.843, respectively, in the validation datasets).

One study (136 patients) focused on gallbladder polyps [50]. It reported that the polyps with a higher skewness (cut-off value = 0.24) and a lower GLCM contrast (cut-off value = 24.6), evaluated with percutaneous US, had an increased risk of adenocarcinoma. The presence of at least one radiomic feature led to a diagnostic accuracy of 0.808. When combined with clinical data (sessile polyp and polyp size), the accuracy increased to 0.885–0.910.

## 4. Discussion

During the most recent years, the radiomics of the biliary tumors, mainly of ICC, has been the subject of major research. As previously reported for other neoplasms [16,18,19,20], textural features had an association with both pathology data and survival and allowed for their accurate prediction. The combination of clinical and radiomic data into a single predictive/prognostic model achieved the best performances, outperforming those of pure clinical models. Radiomics could also contribute to the differential diagnosis of ICC from HCC, mixed ICC–HCC, and inflammatory masses. Notwithstanding this, the heterogeneity of studies (imaging modalities, phases, and software) and the adoption of radiomic signatures/scores preclude the immediate applicability of results to the current clinical practice.

The strength of radiomics relies on its capability to easily extract pixel and voxel patterns from standard imaging modalities, which may predict tumor biology and prognosis [17,51]. Several studies regarding radiomics in oncology have been published, but, unfortunately, the quality of evidence was very low in most cases [19,20,52]. For biliary tumors, the present review outlined a more favorable scenario. Even if all the 27 papers were retrospective, some merits can be highlighted. First, the QUADAS score demonstrated a low risk of bias. Second, the analyzed studies collected a large number of patients affected by rare diseases (≥100 patients in 16 studies). Third, a publication bias cannot be excluded, but negative results were also reported [26,42]. Fourth, several analyses provided a validation of the proposed predictive/prognostic model, even an external one in some cases [39,46,49]. Finally, ten authors demonstrated a clinical net benefit derived from the application of radiomics to their cohorts [25,32,34,35,38,39,43,46,48,49].

Radiomics provided a reliable prediction of pathology data and survival in most studies [25,26,27,28,29,30,34,35,38,40,41,43,45,47,48,49]. Evidence mainly concerned ICC. Both CT- and MRI-based analyses had good performances, and the most informative phases were the arterial and portal phrases of the two imaging modalities and the diffusion-weighted imaging of MRI. The heterogeneity of analyzed imaging modalities and textural features limits the possibility of drawing more specific conclusions, but two additional data deserve consideration. Firstly, among the first-order features, entropy, kurtosis, and skewness were the most relevant and commonly reported features [25,29,34,42,43,45,47,48,49,50]. Those results are in line with the literature: in many tumors, entropy and kurtosis have been associated with tumor aggressiveness [20,53,54]. Entropy reflects the information content of a given area/volume, i.e., the complexity of values’ distribution in the region of interest. Kurtosis represents the spread of the Hounsfield values around the median, and a greater kurtosis indicates a larger spread. Skewness measures the asymmetry of the distribution of the voxel values. Such parameters depict the tumor heterogeneity and presence of necrotic and hyper-vascularized areas, which in turn, correlate with ICC aggressiveness [55,56]. In fact, entropy, kurtosis, and skewness predicted all relevant aspects of the biology of biliary tumors, i.e., tumor grading [29], lymph node metastases [25,48], microscopic vascular invasion [47], and overall and progression-free survival [25,34,42,45,48,49]. Secondly, several studies demonstrated that the combination of clinical data with radiomic features achieved the best predictive/prognostic performances, outperforming the clinical-only and radiomic-only models [25,28,34,35,43,45,48,49]. The textural features integrate and do not replace clinical data, probably because the two convey different information.

The capability of radiomics to predict tumor behavior is clinically relevant because biliary tumors are aggressive diseases with few non-invasive biomarkers [1,2]. To date, the selection of candidates for surgery relies on morphological criteria (i.e., number, size, and pattern of tumors) and tumor markers’ values, which have a limited association with tumor biology [3,15]. The most relevant prognosticators can be assessed only a posteriori in the surgical specimen (e.g., microvascular invasion, tumor grading…), precluding the possibility of defining a priori the best therapeutic strategy [4,13,14]. Radiomics has the potential to refine and anticipate prognosis estimation and, consequently, personalize treatment planning. In the near future, the implementation of AI-based software will ease the implementation of radiomic analyses into advanced multiparameter predictive models, fully exploiting their potential.

The clinicians could also benefit from the contribution of radiomics for the differential diagnosis of liver masses. Even if radiological hallmarks of HCC and ICC were widely depicted [57,58,59,60], a biopsy is still recommended for all hepatic tumors except for those with a typical HCC pattern (wash-in/wash-out) arising in cirrhotic patients under surveillance [8]. Even this exception has been recently questioned by the evidence of some ICC with enhancement patterns identical to HCC [9,10,61]. The radiomic analysis of liver tumors was able not only to distinguish HCC from ICC with adequate performances, which were excellent when the textural features were combined with the clinical data, but also to identify mixed ICC–HCC forms [31,32,33,36]. Lewis et al. reported that tumor radiomics may improve the discrimination capability of LI-RADS score [33]. A standardized non-invasive radiomic-based diagnosis is fascinating and should be pursued in future research. Textural features have been also used to distinguish ICC from inflammatory masses [39,46]. Such patients are difficult to manage because a liver biopsy cannot completely exclude small neoplastic foci, and treatment often requires major resections with a non-negligible risk [62]. Radiomics had excellent reliability in distinguishing the two (ICC vs. inflammatory masses), especially when textural and clinical data were combined (AUC > 0.85) [39,46].

Biliary tumors are often associated with biliary alterations that could influence radiomic analyses. ICC may infiltrate intrahepatic ducts and lead to peripheral biliary dilation. In the case of large heterogeneous tumors, dilation could be difficult to distinguish from the neoplasm and could be mistakenly included in the segmented region of interest. The limit between the tumor and biliary dilation is not an issue for peri-hilar cholangiocarcinoma and EBDT, but the segmentation of tumors with an infiltrative pattern along the bile ducts or associated with a severe inflammation could be difficult. Of course, only the imaging modalities performed before any biliary drainage can be considered. In patients with multiple stenoses (e.g., those with sclerosing cholangitis), the identification of the target lesion is also difficult. The presence of stones could be an additional confounder. The role of radiomics in such patients requires specific evaluations, even if, as mentioned previously, two preliminary studies reported a good capability of textural features to distinguish inflammatory stenoses, intrahepatic lithiasis, and ICC [39,46]. It is certain that such a complex disease presentation requires segmentation by expert radiologists and likely precludes the adoption of unsupervised automatic segmentation protocols.

The available studies have some limitations. The term “biliary tumors” encompasses heterogeneous diseases (intrahepatic cholangiocarcinoma; peri-hilar cholangiocarcinoma; gallbladder cancer; and distal bile duct cancer), but only a few papers focused on neoplasms other than ICC (two papers analyzed peri-hilar cholangiocarcinoma, two extrahepatic tumors, one gallbladder polyps, and one mixed biliary tumors) [27,30,43,45,48,50]. Furthermore, they provided a fragmented picture: three studies demonstrated the association between radiomics and pathology data in EBDT [27,30,48], while two reported a prognostic role of textural features in peri-hilar tumors [43,45]. Additional analyses are needed to fill this gap and to investigate the specific unmet needs associated with the different tumors.

Two additional limitations can be outlined. First, most authors adopted a radiomic signature or score combining multiple textural features. This approach optimized the predictive/prognostic contribution of radiomics but limited the interpretability and reproducibility of data. Second, the textural features were not standardized across studies and varied according to the adopted software, being an in-house software in most papers. Finally, different studies analyzed different imaging modalities and phases, precluding any comparison and meta-analysis of data.

## 5. Conclusions

Radiomics may have a relevant role in the management of biliary tumors, from diagnosis to treatment planning, especially for ICC. Texture analysis provides new non-invasive biomarkers which are complementary to the standard clinical ones. Future research should address the current limitations of the studies to allow the implementation of radiomics in clinical practice.

## Figures and Tables

**Figure 1 diagnostics-12-00826-f001:**
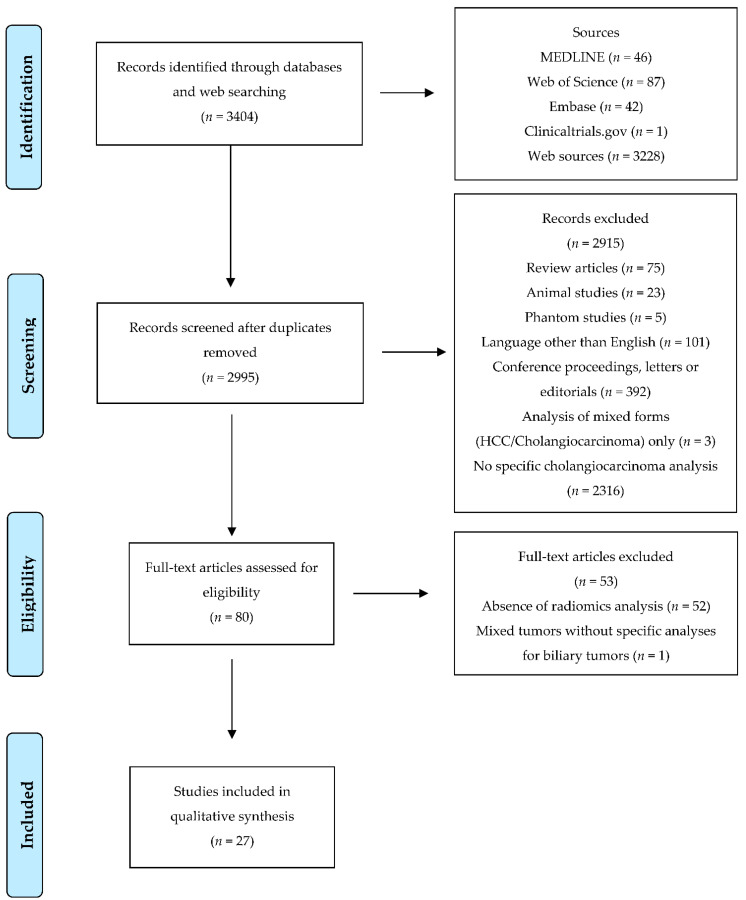
Prisma flowchart.

**Table 1 diagnostics-12-00826-t001:** Details of the analyzed studies.

#	Author	Year	Patients	Tumor Site	Imaging Modality	Analyzed Series	Second-Order Features	Study Design	Validation	Surgery	Analyzed Outcome		
											Diagnosis	Pathology	Prognosis
1	Sadot E [24]	2015	25	ICC	CT	PVP	Y	R	N	N	N	Y	N
2	Choi TW [50]	2018	136	GB polyps	US	-	Y	R	N	Y	Y	N	N
3	Liang W [35]	2018	209	ICC	MRI	AP	Y	R	Y	Y	N	N	Y
4	Xu L [38]	2019	148	ICC	MRI	T1-weighted contrast-enhanced	Y	R	Y	Y	N	Y	N
5	Xu L [41]	2019	332	ICC	CT/MRI	NR	Y	R	Y	Y	N	Y	Y
6	Ji GW [48]	2019	274	Mixed	CT	PVP	Y	R	Y	Y	N	Y	Y
7	Lewis S [33]	2019	17	ICC	MRI	DWI	N	R	N	N	Y	N	N
8	Zhao L [34]	2019	47	ICC	MRI	T2 fat suppr, AP, PVP, LP	Y	R	Y	Y	N	N	Y
9	Ji GW [25]	2019	155	ICC	CT	AP	Y	R	Y	Y	N	Y	Y
10	King MJ [26]	2020	73	ICC	CT/MRI	CT: AP, PVP MRI: AP, PVP, EP, LP, DWI	Y	R	N	Y	N	Y	Y
11	Yang C [27]	2020	100	EBDT	MRI	T1WI, T2WI, DWI	Y	R	Y	Y	N	Y	N
12	Zhang J [28]	2020	98	ICC	MRI	AP, PVP	Y	R	Y	Y	N	Y	Y
13	Peng Y [29]	2020	128	ICC	US	-	Y	R	Y	Y	N	Y	N
14	Yao X [30]	2020	110	EBDT	MRI	T1WI, T2WI, DWI, ADC	Y	R	Y	Y	N	Y	N
15	Zhang J [32]	2020	123	ICC	CT	AP, PVP	Y	R	Y	Y	Y	N	N
16	Peng Y [36]	2020	89	ICC	US	-	Y	R	Y	Y	Y	N	N
17	Mosconi C [42]	2020	55	ICC	CT	AP, PVP, LP	Y	R	N	N	N	N	Y
18	Zhang J [40]	2021	78	ICC	MRI	AP, PVP, unenhanced T1W1, T2WI, DWI	Y	R	N	Y	N	Y	N
19	Qin H [43]	2021	274	PHCC	CT	AP, PVP, DP, EP	Y	R	Y	Y	N	N	Y
20	Chu H [37]	2021	203	ICC	CT	PVP	Y	R	Y	Y	N	N	Y
21	Liu X [31]	2021	24	ICC	CT/MRI	4 CT phases + 9 MRI phases	Y	R	N	N	Y	N	N
22	Xue B [39]	2021	53	ICC	CT	AP	Y	R	Y	Y	Y	N	N
23	Zhu Y [44]	2021	138	ICC	CT	Basal, AP, PVP, LP	Y	R	Y	Y	N	Y	N
24	Zhao J [45]	2021	184	PHCC	MRI	AP and PVP	Y	R	Y	Y	N	N	Y
25	Xue B [46]	2021	61	ICC	CT	AP and PVP	Y	R	Y	Y	Y	N	N
26	Zhou Y [47]	2021	126	ICC	MRI	T2 fat suppr, T1 in-phase and opposed phase, DWI, basal, AP, PVP, DP	Y	R	Y	Y	N	Y	N
27	Park HJ [49]	2021	345	ICC	CT	AP, PVP	Y	R	Y	Y	N	N	Y

ICC: intrahepatic cholangiocarcinoma; GB: gallbladder; PHCC: peri-hilar cholangiocarcinoma; EBDT: extrahepatic biliary tumor; CT: computed tomography; MRI: magnetic resonance imaging; US: ultrasound; Y: yes; N: no; NR: not reported.

**Table 2 diagnostics-12-00826-t002:** QUADAS-2 score of the analyzed studies.

First Author	Patient Selection Bias Risk	Index Test Bias Risk	Reference Standard Bias Risk	Flow and Timing Bias Risk	Patient Selection Applicability	Index Test Applicability	Reference Standard Applicability
Sadot E, 2015 [24]	L	U	U	L	L	L	L
Choi TW, 2018 [50]	L	L	L	U	L	L	L
Liang W, 2018 [35]	H	L	L	U	L	L	L
Xu L, 2019 [38]	H	L	L	U	L	L	L
Xu L, 2019 [41]	U	U	U	U	L	U	L
Ji GW, 2019 [48]	H	L	L	L	H	L	L
Lewis S, 2019 [33]	H	L	L	H	L	H	L
Zhao L, 2019 [34]	H	L	L	U	L	L	L
Ji GW, 2019 [25]	L	L	L	L	L	L	H
King MJ, 2020 [26]	H	L	L	H	L	H	L
Yang C, 2020 [27]	L	L	L	U	L	L	L
Zhang J, 2020 [28]	H	U	U	L	L	L	L
Peng Y, 2020 [29]	H	U	U	L	L	H	L
Yao X, 2020 [30]	L	L	L	L	L	L	L
Zhang J, 2020 [32]	H	L	L	U	L	L	L
Peng Y, 2020 [36]	H	U	U	L	L	H	L
Mosconi C, 2020 [42]	L	L	L	L	H	L	L
Zhang J, 2021 [40]	H	U	U	L	L	L	L
Qin H, 2021 [43]	U	L	L	L	H	L	L
Chu H, 2021 [37]	H	U	U	L	L	L	L
Liu X, 2021 [31]	L	L	L	L	L	H	L
Xue B, 2020 [39]	L	L	L	U	L	U	L
Zhu Y, 2021 [44]	H	L	L	L	L	L	L
Zhao J, 2021 [45]	L	L	L	L	L	L	L
Xue B, 2021 [46]	L	L	L	L	L	L	L
Zhou Y, 2021 [47]	H	U	U	L	H	L	L
Park HJ, 2021 [49]	H	L	L	U	L	L	L

Risk of bias: U, unclear; L, low; H, high.

**Table 3 diagnostics-12-00826-t003:** Pathology data: lymph node metastases, tumor grading and microscopic vascular invasion.

		Radiomic model				Combined Model		
Author *Diagnosis*	Imaging N° Patients	Radiomic Features	AUC (95% CI) Training/Validation	Accuracy (95% CI) Training/Validation	Variables		AUC (95% CI) Training/Validation	Accuracy (95% CI) Training/Validation
**Lymph Node Metastases**								
Ji GW, 2019 [25] *ICC*	CT *N* = 155	Score (kurtosis, GLDM_SDE, GLCM_Contrast, RLNU, and GLNU)	0.823 (0.739–0.907)/0.871 (0.775–0.968)		Radiomics + Ca19-9 + N status at CT		0.846 (0.768–0.925)/0.892 (0.810–0.975)	
Yang C, 2020 [27] *EBDT*	MRI *N* = 100	Signature (no details)	0.880/0.900	0.814/0.833				
Yao X, 2020 [30] *EBDT*	MRI *N* = 110	Signature (no details)	0.904/0.889	0.836/0.812				
Xu L, 2019 [38] *ICC*	MRI *N* = 148	Score (GLCM, GLSZM wavelet transforms)	0.788 (0.698–0.862)/0.787 (0.634–0.898)	0.736/0.691	Radiomics + Ca19-9 + N status at MRI		0.842 (0.758–0.906)/0.870 (0.730–0.953)	0.726/0.786
Xu L, 2019 [41] *ICC*	CT/MRI *N* = 332	Signature (no significant features)	0.704/0.729					
Ji GW, 2019 [48] *Mixed BT*	CT *N* = 274	Signature (shape_MinorAxis, firstorder_Skewness, glszm_ZoneEntropy)	0.790 (0.730–0.860)/0.770 (0.660–0.880)		Radiomics + N status at CT		0.810 (0.750–0.870)/0.800 (0.700–0.900)	
**Tumor grading**								
King MJ, 2020 [26] *ICC*	CT/MRI *N* = 73	No association						
Yang C, 2020 [27] *EBDT*	MRI *N* = 100	Signature (no details)	0.780/0.800	0.699/0.710				
Peng Y 2020 [29] *ICC*	US *N* = 128	Score (kurtosis, skewness)	0.732/0.712	0.735/0.722				
Yao X, 2020 [30] *EBDT*	MRI *N* = 110	Signature (no details)	0.891/0.846	0.826/0.809				
**Microscopic vascular invasion**								
Peng Y, 2020 [29] *ICC*	US *N* = 128	Score (IQRs)	0.699/0.756	0.848/0.684				
Zhou Y, 2021 [47] *ICC*	MRI *N* = 126	Score (GLDM, ZP, GLRLM, skewness, and mean)	0.873 (0.796–0.950)/0.850 (0.709–0.991)	0.863/0.868				

ICC: intrahepatic cholangiocarcinoma; EBDT: extrahepatic biliary tumor; BT: biliary tumors; CT: computed tomography; MRI: magnetic resonance imaging; US: ultrasound; AUC: area under the curve; CI: confidence intervals.

**Table 4 diagnostics-12-00826-t004:** Survival data: overall and recurrence-free survival.

		Radiomics			Combined Model			
Author *Diagnosis*	Imaging N° pts	Signature/Parameter	C-Index (95% CI) Training/Validation	HR (95% CI)	Data	Details	C-Index (95% CI) Training/Validation	Comparison
**Overall Survival**								
Ji GW, 2019 [25] *ICC*	CT *N* = 155	Score (kurtosis, GLDM_SDE, GLCM_Contrast, RLNU, GLNU)		3.650 (1.950–6.830)				
King MJ,2020 [26] *ICC*	CT/MRI *N* = 73	Measure of correlation and ADCmin	NR	NR				
Zhang J, 2020 [28] *ICC*	MRI *N* = 98	AP: LRE, LRHGE, LRLGE, RLNU, SRE	0.673	3.721 (2.210–6.265)	Radiomics + Clinical	Imaging classification (Parenchymal/ductal) CEA	0.721 (0.658–0.783)	Combined better than radiomic and pathology (PD-1, PD-L1, CEA)
Zhang J, 2021 [40] *ICC*	MRI *N* = 78	AP: wavelet-HLH_firstorder_Median	0.700 (0.570–0.820)					
Mosconi C, 2020 [42] *ICC*	CT *N* = 55	No association						
Zhao J, 2021 [45] *PHCC*	MRI *N* = 184	Score: AP (Kurtosis, Correlation, Homogeneity, GLNU, HGRE, Surfac/Volume) PVP (Correlation, SRHGE)	0.877 (0.774–0.979)/0.756 (0.615–0.897)		Radiomics + Clinical Radiological	CEA, N stage at imaging, invasion of hepatic artery at imaging	0.962 (0.905–1)/ 0.814 (0.569–1)	Combined better than clinical and radiomic (clinical similar to radiomic)
Ji GW, 2019 [48] *Mixed BT*	CT *N* = 274	Score (GLSZM_Zone Entropy, Skewness, Minor Axis)		3.370 (1.920–5.910)				
Park HJ, 2021 [49] *ICC*	CT *N* = 345	Score (GLCM_Entropy, GLDZM_HGE, Mean)			Radiomics + Clinical Radiological	Tumor contour, multiplicity, periductal tumor infiltration, extrahepatic organ invasion, suspicion of LN metastases	0.750/0.680	Combined better than clinical-radiological
**Recurrence-free Survival**								
Ji GW, 2019 [25] *ICC*	CT *N* = 155	Score (kurtosis, GLDM_SDE, GLCM_Contrast, RLNU, GLNU		2.770 (1.580–4.840)				
King MJ,2020 [26] *ICC*	CT/MRI *N* = 73	MRI variance		0.550 (0.310–0.970)				
Mosconi C, 2020 [42] *ICC*	CT *N* = 55	Signature (mean, kurtosis, skewness, GLCM_Homogeneity, GLCM_Dissimilarity)		0.460 (0.220–0.950)				
Ji GW, 2019 [48] *Mixed BT*	CT *N* = 274	Score (GLSZM_Zone Entropy, Skewness, Minor Axis)		1.980 (1.260–3.120)				
Park HJ, 2021 [49] *ICC*	CT *N* = 345	Score (GLCM_Entropy, GLDZM_HGE, Mean)	0.690 (0.660–0.750)/0.680 (0.610–0.740)		Radiomics + Clinical Radiological	Tumor contour, multiplicity, periductal tumor infiltration, extrahepatic organ invasion, suspicion of LN metastases	0.750 (0.720–0.790)/0.710 (0.640–0.770)	Combined better than radiomic and clinical-radiologic. Clinical-radiological similar to radiomic

ICC: intrahepatic cholangiocarcinoma; PHCC: peri-hilar cholangiocarcinoma; BT: biliary tumors; CT: Computed tomography; MRI: Magnetic resonance imaging; Pts: patients; HR: hazard ratio; CI: confidence intervals; NR: not reported; AP: arterial phase; LP: late phase.

**Table 5 diagnostics-12-00826-t005:** Early recurrence.

Author *Diagnosis*	Imaging N° pts	Radiomics			Combined Model			
		Signature/Parameter	AUC (95% CI) Training/Validation	Accuracy (95% CI) Training/Validation	Data	AUC (95% CI) Training/Validation	Accuracy (95% CI) Training/Validation	Comparison
Zhao L, 2019 [34] *ICC*	MRI *N* = 47	Radiomic model (AP skewness; PVP variance; AP_Cluster-Shade_AllDirection_offset7_SD; AP_GLCMEntropy_angle45_offset7)	0.889 (0.783–0.996)	0.809	Radiomics + Enhancement pattern, VEGFR	0.949 (0.894–1.000)	0.872 (0.743–0.952)	Combined better than clinical, radiomics and pathological models
Liang W, 2018 [35] *ICC*	MRI *N* = 209	Radiomics score (LRE, HGZE, Mean, GLCM_energy, and SZE)	0.820 (0.740–0.880)/ 0.750 (0.680–0.810) * 0.770 (0.650–0.860) **		Radiomics + TNM	0.900 (0.830–0.940)/ 0.840 (0.780–0.890) * 0.860 (0.760–0.930) **		Combined better than radiomic model
Xu L, 2019 [41] *ICC*	CT/MRI *N* = 332	Score (not detailed)	0.742 (0.666–0.809)/ 0.789 (0.655/0.889)	0.749/0.743				
Qin H, 2021 [43] *PHCC*	CT *N* = 274	AP: S(3,3)AngScMom, motion_S(5,5)SumEntrp, disk_135dr_RLNonUni PP: S(5,-5)SumEntrp, motion_S(2,-2)SumEntrp, motion_S(0,4)Correlat VP: motion_S(0,5)DifVarnc, motion_S(2,-2)DifVarnc, disk_S(3,3)Entropy	0.805/ 0.719 * 0.714 **	0.737 0.671 * 0.649 **	Radiomics + Grading, N status, CA 19-9, Enhancement pattern	0.883/ 0.867 * 0.861 **	0.826/ 0.757 * 0.757 **	Combined better than clinical and radiomic models

ICC: intrahepatic cholangiocarcinoma; PHCC: peri-hilar cholangiocarcinoma; CT: computed tomography; MRI: magnetic resonance imaging Pts: patients; * internal validation; ** external validation; AP: arterial phase; PP: portal phase; VP: venous phase; AUC: area under the curve; CI: confidence intervals.

## Data Availability

Data are available and can be obtained from the corresponding author upon reasonable request.

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
