# Peer review of "Radiomics of Biliary Tumors: A Systematic Review of Current Evidence"

_diagnostics, 2022, doi:10.3390/diagnostics12040826_

Round 1

Reviewer 1 Report

Comments for the Authors:

The manuscript titled “Radiomics of biliary tumors. A systematic review of current evidence” report an extensive review of the literature of an emerging and interesting topic. The systematic review was conducted according to the correct methodology/guidelines and the manuscript is well written.  

Some comments should be addressed to the Authors:

  1. In the methods, according to the PRISMA flowchart (figure 1), the Authors exclude three study regarding HCC/ICC mixed forms, however included the study of Zhang [29], that analyzed both ICC and HCC/ICC. Consider to excluded also this study because could led to confounding results and misinterpretations.
  2. Given that the biliary tumors represents a heterogenous group of malignancies a specific paragraph of the impact of radiomic according the different subtypes (ICC, PCC, EBDT) should be added in results or in the discussion  

Reviewer 2 Report

The review summarizes the possibilities of the radiomics approach for clarifying the diagnosis and prognosis of hepatic and biliary tumors. Twenty-eight studies including 3160 patients were chosen and analyzed. A meaning of texture analysis for three main outcome measures was demonstrated: prediction of pathology data; prediction of survival; differential diagnosis. The review is written clearly and presents new and interesting information on the subject under consideration. The only remark I would like to make concerns the discussion. Commentary on texture characteristics that have been found to be informative in the analysis of various medical images is limited to one sentence, i.e., in fact, their enumeration:

Firstly, among the first-order features, entropy, kurtosis, and skewness were the most relevant and commonly reported ones [24,28,34,42,43,45,47-50]. Those results are in line with the literature: in many tumors, entropy and kurtosis have been associated with tumor aggressiveness [19,53,54].

Since these characteristics and the models formed on their basis can serve as a prognostic / predictive criterion in this category of patients, they deserve a more detailed and in-depth discussion.

Reviewer 3 Report

Biliary tumors (known as cholangiocarcinoma) are heterogeneous and rare diseases (with an incidence of 1-2 people per 100.000 inhabitants in the Western World), with a poor prognosis, poor overall survival, and unsolved issues linked to the impossibility of standard imaging to provide neither a conclusive diagnosis nor robust biomarkers to drive treatment planning. Radiomics, instead, represents one of the most promising fields of cancer research. Through artificial intelligence (AI) and machine learning (ML) advanced algorithms, it is now possible to extract abundant quantitative features from patients scans and to analyze the high amount of data coming from these novel diagnostic tools to ultimately improve the cancer detection rate and its management.

Indeed, this study aims to summarize the available evidence about the role of radiomics in the management of biliary tumors. Thanks to some common characteristics of this heterogeneous neoplasm, a systematic review was carried out and three main outcomes were evaluated: prediction of pathology data, prediction of survival, and differential diagnosis.

The authors included 28 retrospective studies, including 3160 subjects, before April 30th, 2021.

I believe that the study has sufficient merit to be considered for publication, although major revisions are required. However, the manuscript is well-written, easily readable, tables and graphics are clearly described, but it is lacking in some points that would add value to the entire manuscript:

Introduction.

Referred to line 48: authors should provide more information about this common characteristic of biliary tumors. Are they clinical, surgical, or treatment-related features?

Indeed, the paper could take advantage of the description of the main differences between these tumors.

Referred to lines 57-59: are there articles that support these data?

Referred to lines 63-69: A description of the radiomic method misses. This article (https://doi.org/10.3390/ijms22189971) offers an accurate overview. The paper could benefit from its deepening.

The methods and methodology were robust.

Discussion.

The authors should also discuss if and how the presence of coexisting biliary comorbidities could affect the accuracy of these methods.

Reviewer 4 Report

The authors reviewed studies that analyzed the benefit of radiomics in the clinical management of biliary tumors.

The manuscript is overall well-written, and points summarized in discussion are helpful for readers to get take home messages.

However, I have several suggestions to further improve this review manuscript.

  1. The overall statistics is lacking for the focused diseases. In Introduction, biliary tumors are described as rare tumors with poor prognosis, while no detailed data included. For example, in the case of glioblastoma, which is a rare cancer, occurrence rate is about 3-4 per 100,000 population with a 5-year overall survival of about 10%. Such data may help readers to better follow with the authors’ summaries.
  2. Results are a little superficial (e.g., Section 3.2.3), detailed descriptions of studies are lacking. For the study that authors intend to highlight, more details may be given to the readers instead of listing reference numbers. For example, how many patients were analyzed, what question to answer, what detailed features are selected and included, how good the performance is, what might be the limitation for that study.

Such details may help the conclusion to give deeper insights.

Round 2

Reviewer 3 Report

The paper is now suitable for publication.